# Urinary Tract Infection and Microbiome

**DOI:** 10.3390/diagnostics13111921

**Published:** 2023-05-31

**Authors:** Dong Soo Kim, Jeong Woo Lee

**Affiliations:** Department of Urology, Kyung Hee University College of Medicine, Kyung Hee University Hospital, Seoul 02447, Republic of Korea

**Keywords:** microbiome, urinary tract infection, cystitis, urobiome

## Abstract

Urinary tract infection is one of the most common bacterial infections and can cause major burdens, not only to individuals but also to an entire society. Current knowledge of the microbial communities in the urinary tract has increased exponentially due to next-generation sequencing and expanded quantitative urine culture. We now acknowledge a dynamic urinary tract microbiome that we once thought was sterile. Taxonomic studies have identified the normal core microbiota of the urinary tract, and studies on the changes in microbiome due to sexuality and age have set the foundation for microbiome studies in pathologic states. Urinary tract infection is not only caused by invading uropathogenic bacteria but also by changes to the uromicrobiome milieu, and interactions with other microbial communities can also contribute. Recent studies have provided insights into the pathogenesis of recurrent urinary tract infections and antimicrobial resistance. New therapeutic options for urinary tract infections also show promise; however, further research is needed to fully understand the implications of the urinary microbiome in urinary tract infections.

## 1. Introduction

Urinary tract infections (UTIs) are among the most common bacterial infections worldwide and encompass a broad spectrum of diseases, ranging from uncomplicated cystitis to life-threatening urosepsis [1]. Studies using data from the Global Burden of Disease (GBD) show that in 2019 the absolute number of UTI cases had increased by 60.50% since 1990, to a total of 404.61 million patients. The age-standardized incidence of UTIs was 3.6 times higher for females, and UTI incidences increased with age in teenagers, peaking around 35 years. Mortality and disability-adjusted life years (DALYs) were significantly higher in patients aged 65–75 years in both sexes [2]. Due to their high frequency and severity in complex cases, UTIs pose a significant burden for healthcare systems [1,3]. Repetitive use of antibiotics can lead to multidrug-resistant strains, and the cost of treatment can escalate steeply, especially in patients with comorbidities, catheter-associated infections, and septic shock [4].

In-depth evaluation into the micro-organisms involved in UTI has opened doors to a whole new perception of the human urinary tract. Using novel 16S rRNA rapid next-generation gene sequencing (NGS) and expanded quantitative urine culture (EQUC) we now know that the urine is not sterile [5,6]. Over 100 species from more than 50 genera have been identified [7]. The urinary tract is home to a rich and complex microbial community and changes in its composition are thought to be implicated in diverse urinary tract symptoms and diseases [8,9].

Research on the human microbiome and its influence on individual physiology and pathology has increased in recent years [8,10,11,12,13,14]. The implications of microbiome ecology in healthy and diseased patients are yet to be fully understood but future research promises new diagnostic and therapeutic options that include efforts to preserve and restore the integrity of the urinary microbiome [12,13,15,16,17]. This review focuses on recent studies on the relationship between the urinary microbiome and urinary tract infections. We also focus on the newly proposed therapeutic options for recurrent urinary tract infections and their efficacy.

## 2. Materials and Methods

We performed a non-systematic narrative review of articles published in English from 2000 to November 2022. The search terms were as follows: (“urinary tract infection” OR “recurrent urinary tract infection” OR “cystitis” OR “pyelonephritis”) AND (“microbiome” OR “urinary microbiome” OR “urosome”). We searched related articles on PubMed and manually searched the reference lists of the identified articles to identify additional articles. All relevant scientific work including original research, clinical trials, abstracts, and reviews were included.

### 2.1. Urinary Microbiome

#### 2.1.1. EQUC and NGS Technology

Historically, microorganisms have played mixed roles in human life. They have long been used for nutritional and therapeutic purposes [18]. However, humans have also battled against microbial infections and even weaponized them for war [19]. Since the latter half of the twentieth century, the interest in the role of microbial communities residing in the human body has grown [20] and led to huge multinational projects such as the Human Microbiome Project (HMP) [21] and the Metagenomics of the Human Intestinal Tract (MetaHIT) [22] which sought to identify and analyze the human microbiome. The urinary tract was not considered an area of interest in early research because healthy urine was believed to be sterile [5,6]. However, EQUC and NGS technology have helped detect and identify low-biomass communities that have not been reported for standard urine cultures, and further large metagenomic projects show the possibility of functional characterization [23,24,25,26].

NGS refers to diverse new methods of rapid and cost-effective sequencing of DNA and RNA. NGS methods were used to identify bacteria without cultures. Through polymerase chain reaction (PCR) amplification and sequencing, the hypervariable regions of the 16s rRNA gene is analyzed to distinguish between different bacterial species [26]. Using NGS methods, the bladder was proven to be a non-sterile medium [6].

After the discovering the presence of urinary tract microbiomes, EQUC was developed as way to detect microbiomes that were not found with standard urine culture methods. Standard urine cultures used to detect uropathogens in patients with UTI are usually performed using 1 μL of urine in blood agar plates (BAP) and MacConkey agar plates under aerobic conditions at 35 °C. EQUC uses up to 100 μL of urine in BAP, MacConkey, chocolate, colistin–nalidixic acid (NAC) agars, and also CDC anaerobe BAP agars under anaerobic conditions and different gas mixtures at 35 °C. While standard urine culture detects organism growth higher that 1000 CFU/mL, EQUC can detect growth as low as 10 CFU/mL [26]. This enhanced protocol allows detection of up to 92% of bacteria not seen in standard tests [6]. Different protocols using diverse combinations of medium and cultivating environment (extended spectrum) or a concise combination (streamlined) have been reported showing detections rates much higher than standard methods [27].

#### 2.1.2. Uncovering the Urinary Tract Microbiome

Many studies have put their efforts in primarily taxonomically profiling urinary tract microbiomes. For healthy humans, research has mainly been focused on female participants [6,23,28,29,30]. An early study by Siddiqui et al. in 2011 analyzed midstream urine from eight healthy adult female volunteers with confirmed negative cultures. In their study, 16s rRNA sequencing revealed 45 different genera with over 80% assigned to *Lactobacillus*, *Prevotella*, and *Gardnerella*. Variations between individuals were considerable and no common microbial signature was noticeable [28]. Wolfe et al. also analyzed the urine of adult women with no UTIs. The control group consisted of patients with benign gynecologic conditions reporting no urinary symptoms and the comparison group included patient with pelvic floor dysfunction, such as pelvic organ prolapse or urinary incontinence. Urine samples were collected via clean-catch midstream voiding, transurethral catheterization, and suprapubic aspiration and analyzed using bacterial culture, light microscopy, and polymerase chain reaction (PCR) amplification. The presence of bacteria (both Gram negative and positive) was confirmed even in culture-negative transurethral catharized samples. Taxonomical analysis showed *Lactobacillus* was most frequently detected, with additional findings of *Gardnerella*, *Prevotella*, *Actinobaculum*, *Aerococcus,* and *Streptococcus* [23]. The same group also confirmed that bacterial sequences detected using 16s rRNA sequencing could be grown using EQUC in catheterized urine from women with or without overactive bladder. The most common genera were *Lactobacillus*, *Corynebacterium*, and *Streptococcus* [6]. A more recent study by Price et al. examined catheterized urine from continent adult women without infection or other diseases affecting the lower urinary tract. EQUC, 16s rRNA sequencing, and bioinformatics analysis showed *Lactobacillus*, *Streptococcus*, *Gardnerella,* and *Escherichia* were the dominant bacterial genera. Differences between women were associated with age, menopausal status, vaginal parity, and vaginal intercourse [30].

Sex is a major factor that can contribute to differences in baseline individual microbiomes. Compared to women, only a number studies have evaluated male urinary tract microbiomes and most were conducted in the clinical setting of sexually transmitted infections (STIs) [31,32,33,34]. Nelson et al. used 16s rRNA sequencing on first-catch urine in males visiting an STI clinic without symptoms of urethritis. *Lactobacillus*, *Corynebacteria*, and *Streptococcus* were among the most frequently detected genera. Clustering analysis showed that STI were associated with organisms not usually related to the male urinary tract such as *Sneathia*, *Gemella,* and *Prevotella* [31]. They also compared microbiomes in urine and urethral samples from 32 men. They were found to be nearly identical regardless of inflammation or infection. For non-STI patients *Lactobacillus*, *Sneathia*, and *Veillonella* were most abundant, while for STI patients *Neisseria*, *Streptococcus*, and *Corynebacterium* were most common [32]. A later study by Nelson assessed urine and coronal sulcus samples in asymptomatic male adolescents with varied circumcision status and sexual histories. The bacterial communities of urine and coronal sulcus showed distinct differences; the three most common genera for each specimen were *Streptococcus*, *Lactobacillus*, and *Gardnerella* against *Corynebacteria*, *Staphylococcus*, and *Anaerococcus*. Coronal sulcus microbiomes were strongly influenced by circumcision, and sexual activity seemed to have an impact on differences in urogenital microbiota [33]. Froulund et al. also analyzed first-void urine samples in idiopathic urethritis patients against a control group and found high variations between samples with no genus present in all samples. In the control group, Lactobacillus and an unclassified Alphaproteobacterium were present in 50% and 71% of samples. Fouts et al. used 16s rRNA sequencing and metaproteomics to analyze the urine microbiome in neurogenic bladder patients with spinal cord injury. *Lactobacillus*, *Enterobacteriales*, and *Actinomycetales* were among the most common taxa. *Lactobacillus*, *Corynebacterium*, *Gardnerella*, *Prevotella,* and *Enterococcus* defined sex differences. Healthy male bladder microbiomes were defined by *Corynebacterium* while *Lactobacillus* contributed to those of healthy females [24].

As for differences due to age, Price et al. showed that *Gardnerella* was found more often in younger women (mean age 36 ± 13, *p* < 0.001) and those with fewer median vaginal parities, while women with *Escherichia* were older (mean age 60 ± 13, *p* = 0.005) and more likely to be postmenopausal [30]. Lewis et al. studied microbiota from healthy individuals using 16s rRNA sequencing. Although there were only 6 males included in the study, the numbers of genera were analyzed against age. The number of genera in microbiota increased while the average total number of bacteria decreased [35].

Anatomical differences, different urine collection methods, and hormonal changes are considered the causes of these differences, but further research is needed for confirmation [30,36]. A summary of the studies on urinary tract microbiome is presented in Table 1.

### 2.2. Implications of Urinary Tract Microbiome in UTIs

#### 2.2.1. Urinary Microbiome and UTIs

UTIs can occur when the equilibrium between the host and urinary tract microbiome is disrupted [37,38]. Wilner et al. profiled microbial communities in 50 patients with acute uncomplicated UTIs using 16s rRNA sequencing and *fimH* amplicon pyrosequencing from midstream urine samples. UTI microbial communities showed differences according to age with younger individuals dominated with *Escherichia coli* (*E. coli*) and older patients with high abundances of *Pseudomonas*. In females with UTIs, a relatively higher proportion of *Enterobacteriaceae* was observed while in male UTI patients *Streptococcus*, *Lactobacillus*, and *Staphylococcus* were not detected [39]. Garretto et al. used EQUC and 16s rRNA to analyze the genomes of *E. coli* strains from catheterized urine collected from women with lower urinary tract symptoms, including symptomatic UTIs. The presence of *E. coli* and genomic variations in *E. coli* could not predict UTI symptoms. Urobiome analysis showed *E. coli* was a weak predictor of UTI [40].

Few studies have investigated how the urinary microbiome changes during urinary tract infection. Hasman et al. used whole-genome sequencing (WGS) to evaluate bacteria cultivated from random urine samples of patients with suspected urinary tract infection. WGS aided in identifying cultivated bacteria, especially in polymicrobial samples, and predicted antimicrobial susceptibility. In addition, some putative pathogenic strains were observed in culture-negative samples [41]. Price et al. used EQUC to analyze urine samples in women with or without UTI symptoms [27]. Differences in organism diversity and microbiota composition were observed between the two groups. *E. coli* was more prevalent in women with UTI symptoms, whereas *Gardnerella vaginalis*, *Streptococcus mitis/oralis/pneumoniae*, *Streptococcus parasanguinis*, *Streptococcus salivarius*, and *Streptococcus sanguinis* were found more often in women without UTI symptoms. When the analysis was focused on uropathogen detection, *Klebsiella pneumoniae* and *Streptococcus agalactiae* had a significantly higher average of colony-forming units (CFU) in the UTI group. *Aerococcus urinae*, *Enterococcus faecalis*, *E. coli*, *Staphylococcus aureus*, and *Streptococcus anginosus* also had a higher average of CFU. Moustafa et al. used 16s rDNA and metagenome sequencing to profile the urinary tract microbiome. Three distinctive microbial clusters were observed: non-UTI clusters were dominated by *Actinobacter* and *Firmicutes,* whereas the UTI clusters were dominated by *Proteobacteria* and included many other uropathogens, including *E. Coli*, *Klebsiella*, *Pseudomonas,* and *Enterobacter*. Novel candidates for UTI causes, such as *Alloscardovia* and *Actinotignum* were identified, and Candida species, viruses, and phages were also detected [37].

These studies show that multiple factors are involved in UTIs. The urobiome composition appears to play a greater role in urinary tract defense, and its disequilibrium may contribute to infection as much as the invading uropathogen itself. In addition, the virome [42] and mycobiome [43] can also affect the microbiome and should be explored for their role in calibrating the urinary tract milieu. Interactions between different microbial communities, such as the gut and vagina, are also major contributors to tract infection [44]. Metagenomic sequencing of the female urinary tract and vagina showed similar microbiota, suggesting crosstalk between these communities [45]. Recent analysis of human feces also revealed that strains of *E. coli* from the urine and gut were closely related, supporting the hypothesis of a gut microbiota–UTI axis [46].

#### 2.2.2. Recurrent Urinary Tract Infections and the Effects of Antibiotics on the Microbiome

Recurrent UTIs pose a significant burden on both personal and social aspects, from decreasing the quality of life (QOL) to socio-economic issues [1]. The pathogenesis of recurrent UTIs is yet to be defined; however, two major theories and supporting studies have been proposed for both [47]. First, repeated infections via an ascending fecal–perineal–urethral route are supported by studies that show similarities between urine and gut microbiomes and evidence of an intestinal reservoir for recurrence [46,48]. Magruder et al. used 16s rRNA sequencing and metagenomics on fecal and urine specimens of kidney transplant recipients and found that the gut abundance of *Escherichia* and *Enterococcus* were independent risk factors for bacteriuria. Strain analysis also showed a close strain-level alignment in the gut/urine specimens [46]. Forde et al. analyzed the dynamics of an *E. coli* ST131 population for over 5 years in a woman with recurrent UTIs since 1970. WGS was used to identify a clonal lineage linking recurrent UTIs and fecal flora, providing evidence of an intestinal reservoir [48]. A second theory theorizes that the survival of bacteria in the bladder by intracellular bacterial communities (IBC) progresses into persistent quiescent intracellular reservoirs (QIR). While earlier studies focused on recreating the IBC/QIR cycle in murine models [49], recent studies provide evidence of bacteria residing in the bladder wall of patients with recurrent UTI along with alterations of the urothelial architecture [50,51].

Along with recurrent UTI, the emergence of multidrug resistance (MDR) is a global problem. Diagnostic and antimicrobial stewardship are crucial to avoid unnecessary diagnostic testing and to prevent antibiotic misuse [52,53]. Microbiota are thought to gain MDR by acquiring genotypes through horizontal gene transfer or mutations to resist treatment [44,54]. Antibiotic treatment is still the mainstay for urinary tract infections, and these regimens can cause long-term changes in the microbiome. Gottschick et al. compared the urinary microbiota of healthy participants with those of patients with bacterial vaginosis during infection and after antibiotic treatment [55]. Antibiotic treatment reduced *Gardnerella vaginalis*, *Atopobium vaginae*, and *Sneathia amnii* abundance, while *Lactobacillus iners*, which has a controversial role in bacterial vaginosis, was seen to increase to numbers two times higher than those in healthy women. Antibiotic treatment was unable to restore the infected urinary tract microbiome to its pre-infection status. Mulder et al. used 16s rRNA sequencing in elderly participants and analyzed the results against antimicrobial drug use [56]. Previous antimicrobial uses were associated with different compositions of the genitourinary microbiota. Operational taxonomic units (OTUs) of *Lactobacillus* and *Finegoldia* were lowest in those with antibiotic use, whereas *Parabacteroides* and *E. coli* had higher abundances. Considering the pace of multidrug resistance development against new antibiotic development and the burden of recurrent UTI on public health, molecular studies, and microbiome analysis provide promising new treatment modalities that could supplement existing mainstay treatments.

### 2.3. Microbiome and Emerging Treatments for Recurrent UTIs

Research on recurrent UTIs and antimicrobial resistance using cutting-edge technology has provided new insights and an understanding of how microbial communities interact and change pathological states. This knowledge can translate to the development of urinary tract microbiome modulation as a possible method for UTI treatment.

The effectiveness of protective bacteria, such as *Lactobacilli*, on enhancing the defensive mechanisms of the urinary tract have been widely evaluated [12]. These probiotics, when delivered in sufficient amounts, are thought to benefit the host through antimicrobial components such as bacteriocins, biosurfactants, and lactic acids [10]. *L. crispatus* is one of the well-known strains in maintaining a healthy microbiome. Stapleton et al. performed a randomized double-blind placebo-controlled phase 2 trial for an *L. crispatus* intravaginal suppository probiotic (Lactin-V). A total of 100 premenopausal women with a history of recurrent UTIs, were treated for acute infection and randomized for Lactin-V or a placebo. After 10 weeks, UTI recurrence was seen in 15% of the women receiving Lactin-V against 27% of the placebo group (relative risk 0.5, 95% confidence interval, 0.2–1.2). Higher amounts of vaginal colonization by *L. crispatus* were associated with significant reduction in UTI for the Lactin-V group [57]. A more recent paper from 2020 reported a randomized double-blind placebo-controlled phase 2b trial for Lactin-V against a placebo in 228 women who had undergone treatment for bacterial vaginosis. After 12 weeks, the Lactin-V group showed a statistically lower percentage of patients with bacterial vaginosis compared to the placebo group (30% vs. 45%, risk ratio 0.66%, 95% confidence interval 0.44 to 0.87, *p* = 0.01). No differences in adverse effects were seen between the two groups [58]. Other *Lactobacilli* strains that are considered candidates for therapeutic use included *L. rhamnosus*, *L. reuteri*, *L. acidophilus,* and *L. casei* [10,12,59]. The *E. coli* strain Nissle 1917 has also been studied for its antibacterial function and probiotic properties through secretion of microcins and competition in iron uptake, and clinical studies are still ongoing [60,61]. However, Cochrane reviews on the effects of probiotics for the prevention of recurrent UTI in various groups showed no significant benefits [15,62].

There have also been studies on using different strains of *E. coli* for bacterial interference [63,64]. *E. coli* 83972, first isolated from a young girl with asymptomatic bacteriuria [65], has mutations which decrease its virulence compared to other *E. coli* strains that can cause symptomatic infection [66]. Bacterial interference is thought to occur from competition for nutrients, bacteriocin production, competition for attachment sites, and prevention of biofilm formation [67]. Sunden et al. randomized patients with recurrent lower UTIs due to incomplete bladder emptying from a neurogenic bladder, to be inoculated with *E. coli* 83972 or saline. The time to first UTI occurrence and the number of UTI occurrences with or without *E. coli* bacteriuria were evaluated. Results showed that the median time to first UTI was longer for *E. coli* 83972 bacteriuria (11.3 months vs. 5.7 months, *p* = 0.0129), and fewer UTI episodes were seen (13 vs. 35, *p* = 0.009) [63]. Darouiche et al. evaluated a different strain, *E. coli* HU2117, which is a papGΔ derivate of *E. coli* with loss of P-specific adherence expression. A multicenter randomized, double-blinded, placebo-controlled trial evaluated the effects of *E. coli* HU2117 in patients with a history of recurrent UTIs due to neurogenic bladder after spinal cord injury. A total of 29% of patients who received bladder inoculation with *E. coli* HU2117 showed multiple UTI episodes compared to 70% of the placebo group who were injected with saline. Additionally, the number of episodes of UTIs/patient-year was lower for the treated group (0.50 vs. 1.68, *p* = 0.02) [64]. Due to a small number of trials and test populations, there are insufficient data on the effectiveness of bacterial interference in preventing UTIs in patients with bladder dysfunction [15]. Still, the EAU guidelines comment on using deliberate colonization with an ABU strain such as *E. coli* 83972 as a way of preventing symptomatic episodes in patients with lower urinary tract dysfunction [68].

Fecal microbiota transplantation (FMT) transfers microbiota from a healthy donor to the patient’s intestine, restoring a healthy microbiome constitution [69]. A few studies have shown that FMT has a role in reducing recurrent UTIs [70,71,72]. Tariq et al. identified patients with more than three UTIs in the year preceding FMT for three or more episodes of PCR-positive *C. difficile* infection (CDI). They were compared against a control group with three or more UTI episodes in the year prior to their third CDI. FMT was associated with a significant decrease in UTI frequency, and all patients had complete resolution of CID with no recurrence within a 1-year follow up. Additionally, post-FMT antibiotic-susceptibility profiling of urine cultures showed susceptibility to ciprofloxacin and trimethoprim-sulfamethoxazole which were observed to be resistant in earlier cultures [70]. A case report by Bieh et al. reported a 50-year-old female treated with FMT for recurrent UTIs. She was receiving immunosuppressive treatment due to a kidney transplantation she had received 3 years prior. Despite prophylactic antibiotic treatment with cefpodoxime and cranberry concentrates, eight culture-positive febrile UTIs had occurred within the past 2 years with extended-spectrum beta-lactamase-producing *E. coli* seen in the most recent culture. She received oral frozen capsulized microbiota from a healthy donor and was observed for recurrent infection and any adverse effects. No symptoms or episodes were seen for 9 months after treatment and therefore no additional treatments with antibiotics were needed. Follow-up of urine culture during observation showed a decrease in phylogenetic diversity after an initial increase right after FMT. *Enterobacteriaceae* was seen to decrease over time while relative abundances of bacteria from *Bacilli* and *Clostridia* classes were seen [71]. A more recent case control presented by Aira et al. also reported successful resolution of recurrent UTI after FMT. A 93-year-old female with an end-sigmoid colostomy for acute diverticulitis, chronic hepatitis C, and recurrent UTIs was recommended FMT for recurrent episodes of CDI. She received FMT from two healthy related donors via colonoscopy through her colostomy. Two weeks after treatment the colonoscopy showed resolution of pseudomembranes and no other treatment were needed for CID or UTI during the following year [72]. These studies have led to studies on microbiota transplantation in other niches, such as the vagina or urinary tract [12].

Antimicrobial peptides such as human cathelicidins, defensins, and bacteriocins are secreted from the urothelium and exert direct antibiotic activity as well as immunomodulatory effects. Especially bacteriocins, which are peptides produced by bacteria to eradicate competing organisms, have gained renewed interest as a new target for treatment in this era of growing multidrug resistance [10]. *E. coli* produces two types of bacteriocins; high-molecular-weight colicins and small microcins. Colicins damage target cells by membrane permeabilization through voltage-dependent channels, cellular nuclease degradation, and inhibition of peptidoglycan synthesis [73]. They have been reported as a way to inhibit and reduce biofilm formation on urinary catheters [74,75]. Trautner et al. conducted an in vitro study using segments of urinary catheters inoculated with colicin-producing *E. coli* K-12. Theses catheters were exposed to either colicin-susceptible or resistant *E. coli* and incubated overnight. The presence of colicin was sufficient in inhibiting *E.coli* growth in susceptible strains [74]. Roy et al. also evaluated the ability of colicins in preventing extraluminal catheter contamination during insertion. A colicin-mixed catheter lubricant was created by adding partially purified colicin to sterile lubricant (colicin concentration 2.85 μg/mL). Antibiotic resistance levels of uropathogenic *E. coli* were evaluated and compared against colicin-lubricant use. Colicin use achieved the same antimicrobial efficacy as using gentamycin, but at a 20–30% smaller dosage [75].

Other interesting candidates for treatment are the urinary virome and phages [76]. Studies on the virome community are even scarcer than on microbiomes and current analyses are centered on eukaryotic viruses, namely the human papilloma virus (HPV) [42,77]. However, viruses within the human microbiota far outnumber bacterial cells, with the most abundant viruses being those that infect bacteria (bacteriophages) [78], implicating the existence of a large undiscovered viral community. Natural or genetically engineered phages and their lytic enzymes could be considered alternatives to antibiotic treatment or synergistic additives to conventional treatment. However, due to their high specificity, there is an increased risk of bacteria evolving into phage-resistant strains, and most phage therapies are developed in combination with other antimicrobial agents or as phage cocktails [76]. Leitner et al. recently reported the results of the first randomized, placebo-controlled, double-blind trial investigating bacteriophages in UTI treatment. Patients undergoing transurethral resection of the prostate were enrolled in this study when the presence of a symptomatic, non-febrile, non-systematic UTI with an accompanying positive urine culture of at least 10⁴ colony-forming units was observed. They were randomized into three groups to receive either an intravesical bacteriophage solution, an intravesical placebo solution, or antibiotic treatment according to the culture antibiotic sensitivity. Treatment success rates did not differ between groups and normalization of urine culture was observed in 18% of the bacteriophage group, compared to 28% of the placebo group and 35% of the antibiotic group. Adverse events were more favorable for the phage group with 21% compared to 41% in the placebo group (odds ratio 0·36, 95% confidence interval 0.11–1.17, *p* = 0.089) [79]. Studies using microbiomes for UTI treatment are shown in Table 2. 

## 3. Strengths, Limitations, and Future Perspectives

In this review, we have looked over the recent studies on urinary microbiome and its relationship with urinary tract infections. We believe our manuscript provides a much-needed debriefing of a rapidly growing body of scientific literature. It will serve as a starting point for new researchers and/or a stepping stone for ongoing studies. However, the limitations of this study are also evident. This is a non-systematic review and therefore prone to weaknesses such as a non-systematic literature search, author bias, and lack of quality assessment and transparency. Additionally, the urinary microbiome is a relatively new field of research compared to the gut microbiome due to early biases that the human urine is sterile. Its role in normal human physiology or implications in disease pathology are presumed, but due to a lack of supporting studies, there is a lot of controversy. No standardized evaluation methods (urine collection route, measurement modalities) have been proposed which can cause confusion when comparing study results. There is no evidence proving the cost–benefit ratio of microbiome evaluation in patients with UTI and the clinical benefits of microbiome manipulation are limited to selected cases.

Larger RCT trials and real-world data will be needed to prove the efficacy of novel treatments using the microbiome. Hopefully, future research will be able to shed light on a more definitive solution for recurrent UTIs and multidrug resistance.

## 4. Conclusions

NGS technology, EQUC, and large metagenomic projects have helped us detect and identify the microbiome community of the urinary tract, and we now have a better understanding of the pathogenesis of UTIs and their recurrence. These discoveries will hopefully give mankind new insights on managing recurrent UTIs and a desperately needed boost in catching up on multidrug resistance in uropathogenic bacteria. New treatment modalities that use microbiomes as probiotics or competitive substitutes may help prevent recurrent UTIs. In addition to antimicrobial proteins, bacteriophages show promising results as an adjunctive treatment modality to combat antibiotic resistance.

## Figures and Tables

**Table 1 diagnostics-13-01921-t001:** History of microbiome.

Author/Date	Subjects	Evaluation Method	Taxa	Significance
Female patients				
Siddiqui et al., 2011 [28]	8 healthy females	Clean-catch midstream urine 16s rRNA sequencing	*Lactobacillus, Prevotella, Gardnerella*	Normal female urine shows rich bacterial profile
Wolfe et al., 2012 [23]	Women undergoing gynecologic surgery	Midstream, catheterized, suprapubic aspirated urine Culture, microscopy, 16S rRNA sequencing	*Lactobacillus, Gardnerella, Prevotella*	Regardless of symptoms, bacteria not routinely cultivated could be found in urine
Hilt et al., 2014 [6]	Women with overactive bladder (41) and normal bladder function (24) undergoing gynecologic surgery	Catheterized urine EQUC, 16s rRNA sequencing	*Lactobacillus, Corynebacterium, Streptococcus*	Bacteria detected using 16S rRNA sequencing can be grown with enhanced methods
Price et al., 2020 [30]	Continent adult women (224) without lower urinary tract symptoms	Catheterized urine samples EQUC, 16s rRNA sequencing	*Lactobacillus, Streptococcus, Gardnerella*	Bladder microbiome differs by age and gynecological history Younger females: *Gardnerella*, Older females: *Escherichia*
Male patients				
Nelson et al., 2010 [31]	Men without symptoms of urethral infection (19)	First-catch urine specimens 16s rRNA sequencing	*Lactobacillus, Corynebacterium, Streptococcus*	STI causes differences in male urine microbiome composition STI (+): *Prevotella, Sneathia, Gemella*
Donq et al., 2011 [32]	32 men	First-catch urine and urethral swabs 16s rRNA sequencing	STI (-): *Lactobacillus, Sneathia, Veillonella* STI (+): *Neisseria, Streptococcus, Corynebacterium*	First-catch urine and urethral swab microbiomes were nearly identical regardless of inflammation or infection
Nelson et al., 2012 [33]	18 male adolescents	Urine, coronal sulcus samples 16s rRNA sequencing	Urine: *Streptococcus, Lactobacillus, Gardnerella* Coronal sulcus: *Corynebacteria, Staphylococcus, Anaerococcus*	Urine and coronal sulcus host distinct bacterial communities, sexual behavior can alter the urogenital microbiota
Frolund et al., 2018 [34]	Men with idiopathic urethritis (39) and a control group (46)	First-void urine samples 16s rRNA sequencing	Control: *Lactobacillus, Alphaproteobacterium*	Distribution of genera varied considerably between samples and no genus was present in all samples
Male and female patients				
Fouts et al., 2012 [24]	26 healthy controls and 27 subjects at risk of asymptomatic bacteriuria due to spinal-cord-injury-related neuropathic bladder	Urine samples 16s rRNA sequencing, metaproteomics	*Lactobacillus, Enterobacteriales, Actinomycetales*	Urine microbiomes differ by bladder function, gender, type of bladder catheter utilized, and duration of neuropathic bladder Healthy females: *Lactobacillus* Healthy males: *Corynebacterium*

**Table 2 diagnostics-13-01921-t002:** Novel treatments using microbiome.

Author and Date	Subjects	Study Details	Significance
Probiotics			
Stapleton et al., 2011 [57]	Women with a history of recurrent UTIs treated for UTI.	Randomized, double-blind, placebo-controlled phase 2 trial of *L. crispatus* CTV-05 (Lactin-V)	Lactin-V associated with reduction in recurrent UTIs
Cohen et al., 2020 [58]	Women treated for bacterial vaginosis	Randomized, double-blind, placebo-controlled, phase 2b trial of *L. crispatus* CTV-05 (Lactin-V)	Lactin-V resulted in a significantly lower incidence of recurrence of bacterial vaginosis than placebo at 12 weeks
Tapiainen, T NCT04608851 [60]	Young children	Evaluate efficacy of *E. coli* Nissle in secondary prevention of UTI	Ongoing trial
Bacterial interference			
Sunden et al., 2010 [63]	Patients with recurrent UTIs due to neurogenic bladder	Phase 1: randomized, blind, placebo-controlled trial; outcome: time to the first UTI after establishment of *E. coli* 83972 Phase 2: blinded, observational placebo controlled; outcome: number of UTI	Phase 1: median time to first UTI was longer for *E. coli* 83972 bacteriuria Phase 2: fewer UTI episodes were observed
Darouiche et al., 2011 [64]	Patients with recurrent UTI history due to neurogenic bladder after spinal cord injury	Multicenter randomized control trial Evaluated the effects of *E. coli* HU2117	Bladder inoculation with *E. coli* HU2117 showed fewer multiple UTI episodes compared to placebo Number of episodes of UTI/patient-year was lower for the treated group
Fecal microbiota transplantation (FMT)			
Tariq et al., 2017 [70]	Recurrent UTI cases before FMT for recurrent C. difficile infection (CDI)	Retrospectively analyzed against an untreated control group with a prior history of recurrent UTIs and CDI.	FMT successfully treated recurrent CDI, decreased UTI recurrence, and improved antibiotic susceptibility
Biehl et al., 2018 [71]	Kidney transplant recipient with recurrent UTIs	Case report of FMT	No UTI was observed 9 months after FMT and with no adverse effects
Aira et al., 2021 [72]	93-year-old female with an end-sigmoid colostomy and recurrent UTIs	Case report of FMT for recurrent episodes of CDI	Colonoscopy showed resolution of CDI and no other treatments were needed during follow-up
Bacteriocins			
Trautner et al., 2005 [74]	Urinary catheter	Catheter inoculated with colicin-producing *E. coli* K-12 and exposed to *E. coli*	Colicin inhibited *E. coli* growth in susceptible strains
Roy et al., 2019 [75]	Urinary catheter	Catheter was dipped in colicin-mixed catheter lubricant (purified colicin with sterile lubricant)	Colicin achieved the same antimicrobial efficacy as using gentamycin, at a 20–30% smaller dosage
Bacteriophage therapy			
Leitner et al., 2021 [79]	Patients undergoing transurethral resection of the prostate with symptomatic, non-febrile, non-systematic, culture-positive UTIs	Randomized, placebo-controlled, double-blind trial Received either intravesical bacteriophage solution, intravesical placebo solution, or antibiotic treatment	Treatment success rates did not differ between groups Adverse events were more favorable for the bacteriophage group

## Data Availability

Not applicable.

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
