# Peer review of "Urinary Tract Infection and Microbiome"

_diagnostics, 2023, doi:10.3390/diagnostics13111921_

Round 1

Reviewer 1 Report

The work has been adequately reviewed and it is a well written non systemic review where the topic has been treated. The role of the virome, mycobiome has been mentioned and the differences in the male/ female communities of UTI micobiome highlighted. E.coli was found to be a weak predictor of UTI and current treatment options highlighted.

My question is on Line 138-139 where the age bracket of the younger and older patients have not been specified provided it is in the literature. 

Author Response

Thank you for your dedication in reviewing our article. In response to your comment, the cited reference used Wilcoxon rank sum tests to compare the mean age of the patients with different UTI infection. To better convey the results, the mean age and p-value was added to our manuscript.

Reviewer 2 Report

The non-systematic review article entitled Urinary Tract Infection and Microbiome with the aim of presenting information from recent studies on the relationship between UTI and the microbiome, as well as the proposal of new therapeutic proposals for recurrent UTI. The presented and analyzed information allow us to understand the preponderance that microorganisms have in host health and to avoid colonization by uropathogens. However, revising the manuscript I found some minor errors in the text that must be address by the authors before accepting it.

Citations in the text must precede the punctuation mark.

In line 103 the term non-STI is mentioned for the first but there is no explanation for the abbreviation.

The taxa Lactobacillus, Corynebacterium, Streptococcus must be changed to italics in Table 1 line 3.

The name E. coli must be in italics on lines 300, 307 and 308.

Author Response

Thank you for your detailed review of our manuscript. Please excuse our errors in editing and styling this article. We have gone over the text, taking care to correct according to your feedback. Sexually transmitted infection (STI) was used instead of sexually transmitted disease (STD) for consistency.

Reviewer 3 Report

Urinary microbiome is a novel topic which is increasingly studied by recent papers. 

There are a few aspects that I suggest to be modified in the present manuscript.

1. The introduction section should be arranged. I suggest starting with general data from paragraph two and after that to get close to the subject.

2. The authors should decide where to place the citation indicator. I suggest the "[...]" to be placed before "."

3. In the Material and Methods section the authors should explained if they also included trials, abstracts, congress presentations or reviews, or the included only original researches.

4. EQUC and NGS technology should be explained in a separate chapter.

5 Table 1. and 2. should be better organized. The authors may separate women and man. They can also include columns for dividing collecting technique, the analyze technique, patient's characteristics, identified strains etc...The studies publication year can be also specified. These tables are two crowded and the information is hard to follow.

6. The authors should include a strength and limitations section.

7. The conclusion section should be more concise.

8. Last but not least English spelling needs minor improvements.

Minor English spelling issues.

Author Response

Thank you for taking the time and effort in reviewing our manuscript. Your detailed comments are greatly appreciated, and they have us aided immensely in improving the quality of our review. Below are detailed responses for each of your comments.

1. The original introduction started abruptly by mentioning the human microbiome without any context. We have edited the introduction as your suggestions for a more coherent script and improve its readability.

2. We have edited the references to show consistency. Please forgive our lack of prudence in editing.

3. We have specified the type of article that were reviewed to write this manuscript.

4. A new paragraph was added to describe the NGS and EQUC methods and their historical role in urinary tract microbiome research.

5. As you have commented the tables were difficult to comprehend and hard to read. We have edited the tables according to your suggestions and added more columns for easier accessibility.

6. We have added a paragraph describing the strengths, limitations and future prospects of this review.

7. Our conclusion was edited to express a shorter and more compact overview of the manuscript.

8. The entire manuscript was careful reviewed and edited for clerical errors and care was taken to ensure the text was coherent.

Reviewer 4 Report

Thank you for submitting you work to our Journal.

Your work is impressive and made a good reading for me.

In order to keep the scientific structure of a good paper, I suggest you include a paragraph about the potential limits of your work.

Author Response

Thank you for your kind words and thoughtful insight concerning our manuscript. We are thrilled at your positive feedback and added an additional paragraph containing the strengths and limits of this review as you have suggested.

Reviewer 5 Report

This is a well-written, informative, comprehensive and well-structured, non-systematic narrative review of the literature, aiming to investigate the associations between the urinary tract microbiome and UTIs. I have no comments for revisions to the manuscript. 

Some minor editing is required. 

Author Response

Thank you for your time and effort in reviewing our manuscript. We are thrilled that our work has satisfied your high standards and grateful for your positive feedback.

Round 2

Reviewer 3 Report

All the suggestions and requests were successfully accomplished, so I recommended the present manuscript to be accepted.

English improved